

# Modulation of the gut microbiota by processed food and natural food: evidence from the *Siniperca chuatsi* microbiome

Hongyan Li[1,2,3,*], Shuhui Niu[1,4,*], Houjun Pan[1,2,3], Guangjun Wang[1,2,3], Jun Xie[1,2,3], Jingjing Tian[1,2,3], Kai Zhang[1,2,3], Yun Xia[1,2,3], Zhifei Li[1,2,3], Ermeng Yu[1,2,3], Wenping Xie[1,2,3] and Wangbao Gong[1,2,3]

[1] Pearl River Fisheries Research Institute, Chinese Academy of Fishery Sciences, Guangzhou, China

[2] Hainan Fisheries Innovation Research Institute, Chinese Academy of Fishery Sciences, Sanya, China

[3] Guangdong Provincial Key Laboratory of Aquatic Animal Immunology and Sustainable Aquaculture, Guangzhou, China

[4] National Demonstration Center for Experimental Fisheries Science Education, Shanghai Ocean University, Shanghai, China

* These authors contributed equally to this work.

Corresponding authors
Houjun Pan, phj001@prfri.ac.cn
Wangbao Gong, gwb@prfri.ac.cn

## ABSTRACT

Habitual dietary changes have the potential to induce alterations in the host's gut microbiota. Mandarin fish (*Siniperca chuatsi*), an aquatic vertebrate species with distinct feeding habits, were fed with natural feeds (NF) and artificial feeds (AF) to simulate the effects of natural and processed food consumption on host gut microbiota assemblages. The results showed that the alpha diversity index was reduced in the AF diet treatment, as lower abundance and diversity of the gut microbiota were observed, which could be attributed to the colonized microorganisms of the diet itself and the incorporation of plant-derived proteins or carbohydrates. The β-diversity analysis indicated that the two dietary treatments were associated with distinct bacterial communities. The AF diet had a significantly higher abundance of Bacteroidota and a lower abundance of Actinomycetota, Acidobacteriota, and Chloroflexota compared to the NF group. In addition, Bacteroidota was the biomarker in the gut of mandarin fish from the AF treatment, while Acidobacteriota was distinguished in the NF treatments. Additionally, the increased abundance of Bacteroidota in the AF diet group contributed to the improved fermentation and nutrient assimilation, as supported by the metabolic functional prediction and transcriptome verification. Overall, the present work used the mandarin fish as a vertebrate model to uncover the effects of habitual dietary changes on the evolution of the host microbiota, which may provide potential insights for the substitution of natural foods by processed foods in mammals.

## INTRODUCTION

Food is a critical factor in driving host evolution and modulating gut microbiota assemblages (*Teaford & Ungar, 2000*; *Cuevas-Sierra et al., 2021*). The advent of

industrialization has led to remarkable advances in the food processing industry, encompassing various techniques such as crushing, grinding, extrusion, puffing, heating, and other processes (*Rao, 2009*; *Huebbe & Rimbach, 2020*). These techniques have facilitated the mass production and processing of food, ensuring the year-round availability of food to meet growing human demand. As a result, the type and pattern of food consumed by modern humans has also changed greatly compared to our closest primate relatives, who inevitably consumed natural raw foods (*Zhang & Li, 2018*). Diet is known to provide energy and nutrients to humans, but it also has a profound and rapid effect on the composition of the gut microbiota. Remarkable differences in gut microbiota composition have been observed between individuals from urban-industrialized societies and those from traditional rural societies with different dietary patterns (*De Filippo et al., 2010*). In recent decades, the consumption of processed foods has been strongly linked to some health issues such as the high prevalence of obesity in Western countries, primarily due to its influence on gut bacteria and their metabolism (*Miclotte & Van de Wiele, 2020*). However, beyond the effect of changes in dietary nutrients alone, the potential role of processed foods compared to natural foods in shaping the gut microbiota remains largely unexplored.

Similar to mammals, fish species are one of the most representative vertebrates with evolutionarily unique phylogenies in our biosphere (*Zhang & Li, 2018*), so explorations in fish could provide insights into mammalian mechanisms to some extent. Nowadays, many domesticated animals, including certain fish species, are gradually shifting towards the consumption of processed foods produced by humans, especially in industrializing developing societies. The mandarin fish (*Siniperca chuatsi*), a native freshwater aquatic animal found in China and some regions of the Russian borderlands, exhibits a distinct dietary preference in the wild as it could only accept natural fresh fish baits throughout its life (*Liang, Kiu & Huang, 1998*). In the past, live bait was commonly used in mandarin fish farm culture. However, due to seasonal and regional factors, the stable supply of live bait in quantities and sizes appropriate to the species cannot be fully guaranteed, which hinders the development of large-scale culture. In addition, natural fresh baits, where a large number of microorganisms or parasites colonize, inevitably increase the unexpected mortality mandarin fish under farming conditions (*Chen et al., 2022*). As a result, the high economic value of mandarin fish culture and the development of the feed industry have triggered research on mandarin fish domestication since the 1990s, and it has finally been demonstrated that mandarin fish can successfully consume processed foods such as artificial diets through manipulation (*Liang et al., 2001*; *Li et al., 2017*).

Dietary transition from natural food (fresh baits) to processed food (artificial feeds) can promote large-scale culture, save costs, and ensure the food security of mandarin fish. Numerous studies have investigated the morphological changes, physiological activities, and metabolic responses of mandarin fish following this dietary transition (*Li et al., 2017*; *Shen et al., 2021*). The molecular mechanisms underlying these responses have been investigated using mRNA transcriptome analysis, metabolome analysis and epigenetic landscapes. However, it is worth noting that the gut microbiome has been implicated in host development in aquatic animals, including zebrafish (*Danio rerio*), tilapia (*Oreochromis niloticus*), and Atlantic salmon (*Salmo salar*) (*Giatsis et al., 2015*;

*Dvergeda et al., 2020*; *Xiao et al., 2022*). Thus, the effects of artificial feeds on the gut microbiota of mandarin fish warrant further investigation.

Mandarin fish were employed as a research model to understand the effects of dietary shifts from natural food to processed food. Two experimental diets, the natural feeds (NF) and artificial feeds (AF) diets, were used to represent the natural food and processed food, respectively. A feeding experiment was conducted on mandarin fish to examine the effects of these two diets over a long period of time. In the present study, alterations in bacterial assemblages and host-associated metabolic changes were evaluated in mandarin fish were evaluated in mandarin fish to reveal the impacts of the diet pattern on the host. In addition, the molecular activities underlying the metabolic changes in mandarin fish after the habitual food conversion were investigated by transcriptome analysis. By incorporating evidence from vertebrates, this research sought to unravel the underlying mechanisms linking host responses and the gut microbiota assemblages following a dietary transition from natural to processed foods.

## MATERIALS AND METHODS

### Animal intervention

The present study was conducted in a pond system at the Pearl River Fisheries Research Institute, Chinese Academy of Fishery Sciences (Guangzhou, China). The animal intervention procedure has been detailed in previous research by *Li et al. (2023)* in details. In brief, domesticated mandarin fish were acquired from YuShun Fisheries Company (Qingyuan, China). Two weeks prior to the experiment, the fish were acclimatized to the experimental system. Then, 600 uniform sized mandarin fish with an average weight of 334 g were randomly distributed into six ponds (with a corresponding density of 9.27 kg/m$^3$). Each treatment contained triplicates tanks. The two treatments were exposed to either natural feeds or artificial feeds for a period of 180 days, referred to as the NF and AF treatments, respectively. The NF diet comprised of living dace and carp (purchased from YuShun Fisheries Company, Qingyuan, China), while the AF diet was a commercial diet purchased from Foshan Nanhai Jieda Feed Co., Ltd (Guangdong, China), which is widely used for mandarin fish rearing. Table S1 presents the basic chemical composition of the experimental diets. Mandarin fish were fed to visual satiation throughout the experiments (at 09:00 and 16:00 a day). The survival of the fish was recorded and tended not to be significant during the rearing procedures with survival rates of 96.3% and 97.3% respectively ($p = 0.47$). Regular monitoring was conducted on water quality parameters, including water temperature (28 °C–34 °C), ammonia nitrogen (below 0.5 mg/L), pH between (6.7–7.2), dissolved oxygen (5.7– 6.8 mg/L), and light intensity (2.83–3.31 μmols/ m$^2$ with a photoperiod based on the natural light). There were no significant differences between the above water quality parameters during the whole feeding trial.

### Sampling procedure

After the feeding trial, dilute tricaine methanesulfonate solution (MS-222, Sigma-Adrich, St. Louis, MI, USA) were used for fish anaesthetization. Two fish from each pond were sacrificed by dissection of the abdominal cavity using sterilized scissors (six fish per

treatment). The entire intestine was rapidly dissected, then the intestinal contents were carefully collected and stored in 1.5 mL sterile tubes. Samples were immediately stored at −80 °C for subsequent analysis. To avoid contamination of the intestine, all the sampling instruments were previously sterilized by autoclaving. During the sampling procedures, the sampling tools were disinfected with 75% medical alcohol between samples and flame sterilized. Two of the whole fish bodies from each pond were collected and kept in the refrigerator at −20 °C for nutrient composition analysis. The animal trial was carried out strictly according to the guidelines of the Institutional Animal Care and Use Ethics Committee of the Chinese Academy of Fishery Sciences (No. LAEC-PRFRI-2021-01-01).

## DNA extraction and 16s rRNA gene sequencing

Total genomic DNA extraction was extracted from the intestinal contents of six samples in each group using a NucleoSpin Soil kit (Macherey-Nagel, Dueren, Germany) following the manufacturer's guidelines. Then, two of samples from the same tank were pooled. DNA integrity was monitored by performing agarose gels. DNA concentrations were detected by a Nanodrop 2000c spectrophotometer (Thermo Fisher, Waltham, MA, USA). Specific primers were used to amplify and sequence the V3–V4 regions of the 16S bacterial DNA gene: 338 F, 5′-ACTCCTACGGGAGGCAGCA-3′; 806 R, 5′-GGACTACHVGGGTWTCTAAT-3′. PCR reactions were performed in a 25 μL reaction systems, containing 5 × reaction buffer (5 μL), 5 × GC buffer (5 μL), dNTP (2 μL), Q5 DNA polymerase (0.25 μL), DNA template (2 μL), each primer (10 μM, 1 μL), and ddH$_2$O (up to 25 μL). The amplification program consisted of an initial denaturation phase (98 °C for 2 min), 30 cycles of extension (98 °C for 15 s; 55 °C for 30 s; 72 °C for 30 s), and a final extension (72 °C for 5 min). The amplification products were acquired from 1.8% agarose gels and purified using Agencourt® AMPure XP Beads (A63881, Beckman Coulter Life Sciences, Brea, CA, USA). The amplicons' quality was then measured using a Qubit® 2.0 Fluorometer (Q32866, Invitrogen, Waltham, MA, USA) and an Agilent 2100 Bioanalyzer (G2939AA, Agilent Technologies, La Jolla, CA, USA). Sequencing libraries were built using the TruSeq Nano DNA LT Library Prep Kit for Illumina (San Diego, CA, USA) according to the manufacturer's guidelines. The quantity and quality of library were monitered using Qubit and TapeStation (Agilent Technologies, La Jolla, CA, USA). Finally, the qualified amplifications were sequenced on the Illumina HiSeq 2500 platform. The SILVA (release 132, https://www.arb-silva.de/documentation/release-132/) and NT-16S databases were used for species alignment and annotating. The raw reads generated in this research were archived in the NCBI Sequence Read Archive (SRA) database with accession number PRJNA897818.

## Bioinformatic analyses

Quantitative Insights into Microbial Ecology version 2 (QIIME2) (*Bolyen et al., 2019*) was employed for data processing. Briefly, QIIME cutadapt trim-paired was called to trim the primer fragments of the sequences and discard the sequences that did not match the primers; then DADA2 was called for quality control, denoising, splicing, and chimera removal by QIIME dada2 denoise-paired. The above steps were performed separately for

each library. Then, raw tags were generated with an overlap length greater than 10 bp and a maximum mismatch rate less than 0.2. FLASH (V1.2.7) was used to assign paired-end reads to samples. Sequences underwent analysis UCHIME algorithm and QIIME (*Caporaso et al., 2010*; *Edgar et al., 2011*). Tags that demonstrated effectiveness were subsequently filtered and clustered into Operational Taxonomic Units (OTUs) utilizing the Greengenes database, with an identity threshold set at 97%. The taxonomic annotation of each OTU was performed using the Uparse software (*Edgar, 2013*).

## Diversity and statistical analysis

Alpha diversity indices, compromising of the observed OTUs, Chao1, Simpson, Shannon, Pielou's evenness index (Pielou-e), Faith's pd diversity, and Good's coverage between gut microbial communities, were analyzed using Mothur (v.1.21.1) and a non-parametric Kruskal-Wallis test. Rarefaction curves were obtained by plotting the number of sequenced reads (x-axis) by the number of OTUs found in a community. QIIME 2 assessed beta diversity and clustered it using weighted and unweighted UniFrac distance metrics. Similarities between groupings were visualized using principal coordinate analysis (PCoA) with unweighted and weighted UniFrac distances. The diet-induced common and unique OTUs were identified using a Venn diagram. The total microbial composition of the different treatments was compared using permutational multivariate analysis of variance (PERMANOVA) with unweighted and weighted distances to identify significant differences. The relative abundance of bacterial taxa was analyzed using linear discriminant analysis effect size (LEfSe) to identify any statistically significant differences. The researchers used Phylogenetic Investigation of Communities by Reconstruction of Unobserved States (PICRUSt) to predict the functional profile of the gut microbiota. PICRUSt uses evolutionary modelling to predict metagenomes from 16 S data and a reference genome database. Predicted functional pathways were annotated using the Kyoto Encyclopedia of Genes and Genomes (KEGG) at Level 2. Further functional categories were performed by metagenome Seq analysis at Level 3 to identify metabolic pathways with remarkable differences between treatments.

## Carbohydrate and lipid related metabolism

Based on the functional prediction of the gut microbiota, we further measured the carbohydrate and lipid-related metabolic indicators of mandarin fish to indicate the nutrient assimilation pattern of the two treatments. As previously described by *Li et al. (2019)*, the liver samples, which had been fixed in 4% paraformaldehyde, were sectioned and embedded in Tissue-Tek OCT compound (Sakura Finetek, Tokyo, Japan). They were rapidly frozen in liquid nitrogen-cooled isopentane and then sectioned at 5 μm using a cryostat. The sections were then stained with neutral Oil Red O (Wako Pure Chemicals, Osaka, Japan) to reveal the presence of liver lipid droplets according to a previous study (*Li et al., 2020*). The number of lipid droplets was quantified using Image-Pro Plus 4.1 software by analyzing ten fields of each sample. Crude lipid content in the muscle was measured in a Soxtec system (Soxtec System HT6, Sweden) using diethyl ether as the extraction fluid.

TRizol reagent (Invitrogen, Waltham, MA, USA) was utilized for total RNA extraction from six samples from each group. A mixture of the liver, muscle, and intestinal tissues was prepared for each sample. The quantity and purity of total RNA was assessed using the Bioanalyzer 2100 and RNA 1000 Nano LabChip Kit (Agilent, La Jolla, CA, USA) following the principle of RIN number greater than 7.0. A cDNA library was generated following poly(A) RNA purification and reverse transcription of cleaved RNA fragments according to the mRNA Seq sample preparation kit protocol (Illumina, San Diego, CA, USA). Paired-end sequencing was then performed on an Illumina NovaseqTM 6000 (LC Sciences, Houston, TX, USA) according to the manufacturer's suggestions. The relative expression of genes related to carbohydrate and lipid metabolism was analyzed according to the Log2 fold change from the transcriptome data (SRA accession number: PRJNA897748). Differentially expressed genes identified for carbohydrate and lipid metabolism by KEGG pathway enrichment analysis were clustered in a hierarchical clustering heatmap using R software (v3.1.0; *R Core Team, 2014*).

## RESULTS

### Overview of sequencing data

In the present study, a total of 780,620 sequences were acquired from six gut samples from each group of mandarin fish fed with NF and AF. After applying quality filtering, 731,660 clean sequences (ranging from 87,624 to 194,875) were classified as OTUs (Table S2), with a mean length of 416 bp for the high-quality processed sequences (Fig. S1). The rarefaction curves of the observed OTUs suggest that there is sufficient sequence data to obtain reliable and unbiased assessments of microbial species richness.

### Alpha diversity and richness

Alpha diversity indices, including Chao1, Simpson, Shannon, Pielou-e, Observed OTUs, and Faith's pd, were measured to evaluate the diversity and richness of the microbiota in the faeces of mandarin fish fed with the NF and AF diet. The identification of microbial phylotypes in all gut samples was robust, as indicated by high Good's coverage estimator values ranging from 98% to 99% (Fig. 1). No significant differences were found between the treatments for Simpson and Pielou-e (Fig. 1; $P > 0.05$). The AF group had significantly lower values for Chao1, Shannon, observed OTUs, and Faith's pd index than the NF group in mandarin fish (Fig. 1; $P < 0.05$).

### Beta diversity of gut microbial communities

The rank abundance curve, which shows the distribution of OTU abundance in each sample by ranking the values. The results of the rank abundance curve showed that the NF group had higher abundance than the AF group (Fig. 2A). The differentially observed OTUs in mandarin fish between the two treatments were identified using a Venn diagram (Fig. 2B). A total of 772 OTUs were shared between the two treatments, and a total of 7,375 OTUs and 3,076 OTUs were uniquely identified in the NF and AF treatments, respectively. PCoA of beta diversity was performed to visualize the similarities between the two

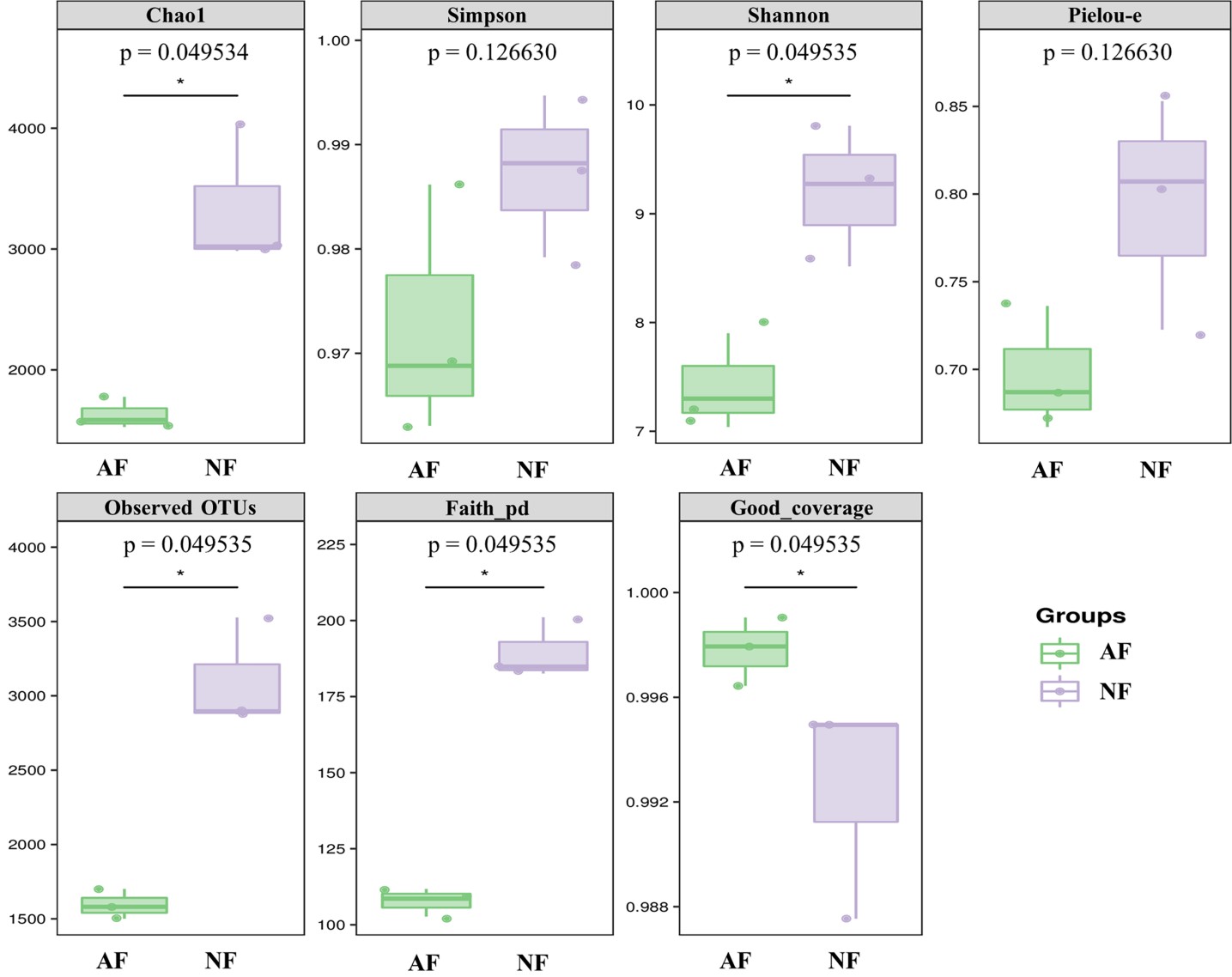

**Figure 1  The alpha-diversity of gut microbial community in mandarin fish fed with the NF and AF diets.** NF, natural feeds; AF, artificial feeds. The asterisk (*) at the top of figures indicates significant differences between the two groups (*P* < 0.05).

treatments with unweighted_unifrac and weighted_unifrac distances (*n* = 3, Figs. 2C, 2D). Both PCoA diagrams showed that samples from the AF group indicated a separation from the NF group, as the microbial communities appeared as two distinct central clusters along with the PCA distance variations on both axes. To provide a clearer visualization of the taxonomic abundance distribution between the two treatments, a heatmap clustering analysis was performed on the 50 most abundant genera, as shown in Fig. S2.

## Differences in taxonomic composition

The bacterial community composition of mandarin fish fed with the NF and AF diets was analyzed at the phyla and genus level (Fig. 3). The top ten most abundant phyla were listed and the remaining phyla were termed 'Others' (Fig. 3A). The predominant phylum

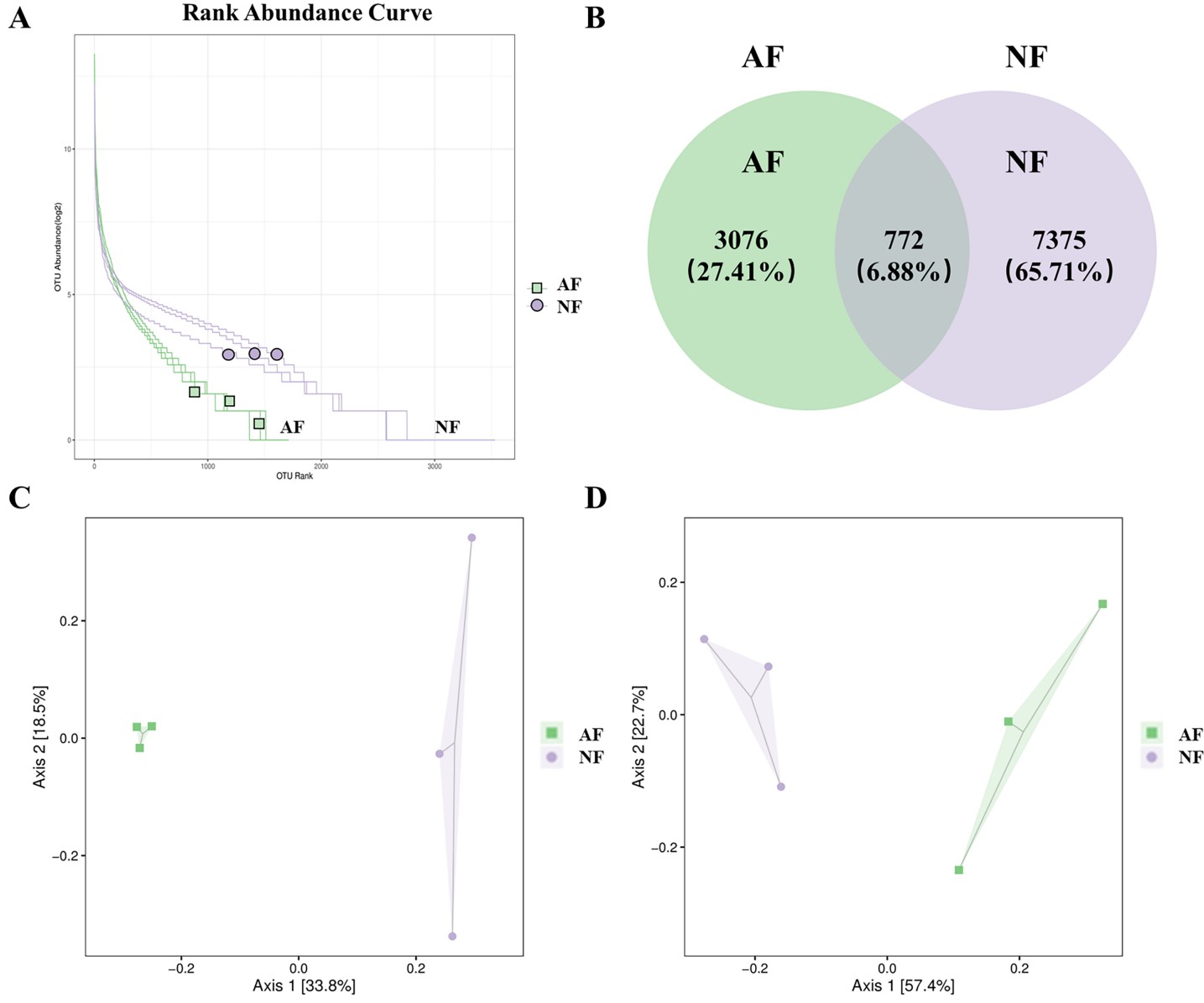

**Figure 2 Distribution of OTUs abundance analysis and β diversity analysis in mandarin fish fed with the NF and AF diets.** NF, natural feeds; AF, artificial feeds. (A) Rank abundance curve. (B) Venn diagram showing the observed OTUs of gut microbial communities. Principal coordinate analysis based on unweighted (C) and weighted UniFrac distances (D) of gut microbiota between groups.

observed in the gut microbiota of both mandarin fish treatments were Bacillota, Pseudomonadota, Actinomycetota, Bacteroidota, followed by Fusobacteriota, Acidobacteriota, Chloroflexota, TM7, Thermi, and Gemmatimonadota. At the genus level, the top ten most abundant taxa were identified as *Lactobacillus*, unidentified *Clostridiales*, unidentified *Peptostreptococcaceae*, *Ruminococcus*, unidentified *Ruminococcaceae*, *Cetobacterium*, *Clostridium*, unidentified *S24-7*, *Lachnospiraceae*, and *Oscillospira* (Fig. 3B).

 

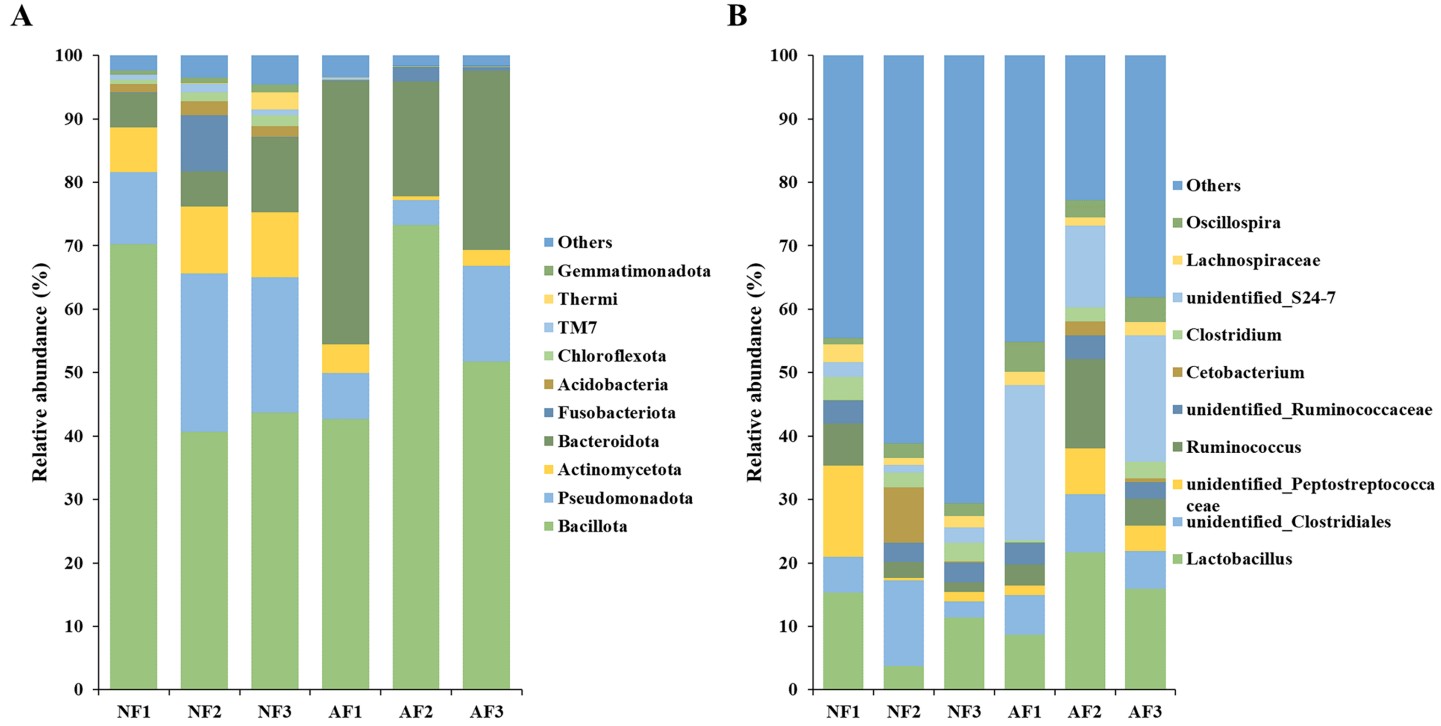

**Figure 3 Taxonomic composition of the gut microbial community at the phylum (A) and genus (B) level in mandarin fish fed with the NF and AF diets.** NF, natural feeds; AF, artificial feeds.

In addition, the abundance of the dominant microbial taxa varied significantly between the two treatments (Fig. 4). In the NF group, Actinomycetota, Acidobacteriota, Chloroflexota, TM7, and Gemmatimonadota exhibited significantly higher relative abundances compared to the AF group of mandarin fish ($P < 0.05$). On the contrary, Bacteroidota showed a significantly higher relative abundance in the mandarin fish fed with the AF diet, compared to the NF group ($P < 0.05$). LEfSe was conducted to further analyze the differential bacterial taxonomic abundances between the two treatments (Fig. 5). As revealed by the LEfSe LDA score (LDA > 4), eight and four bacterial taxa contributed to the AF and NF groups, respectively (Fig. 5A). Additionally, the LEfSe cladogram indicated Bacteroidota as a biomarker in mandarin fish fed with the AF diet, compared to the NF treatments (Fig. 5B).

## Metabolic functional prediction

At Level 2 of the KEGG pathway, a total of six pathways were annotated based on the KEGG database using PICRUSt (Fig. 6A). The predicted pathways are dominated by metabolism, genetic information processing, and cellular process. Further functional categories by metagenome Seq analysis showed significant differences in pathway enrichment between the NF and AF treatments of mandarin fish (Fig. 6B). Compared to the NF group, the AF group showed significantly higher metabolic levels related to pathways such as the PPAR signaling pathway and steroid biosynthesis. In contrast, higher

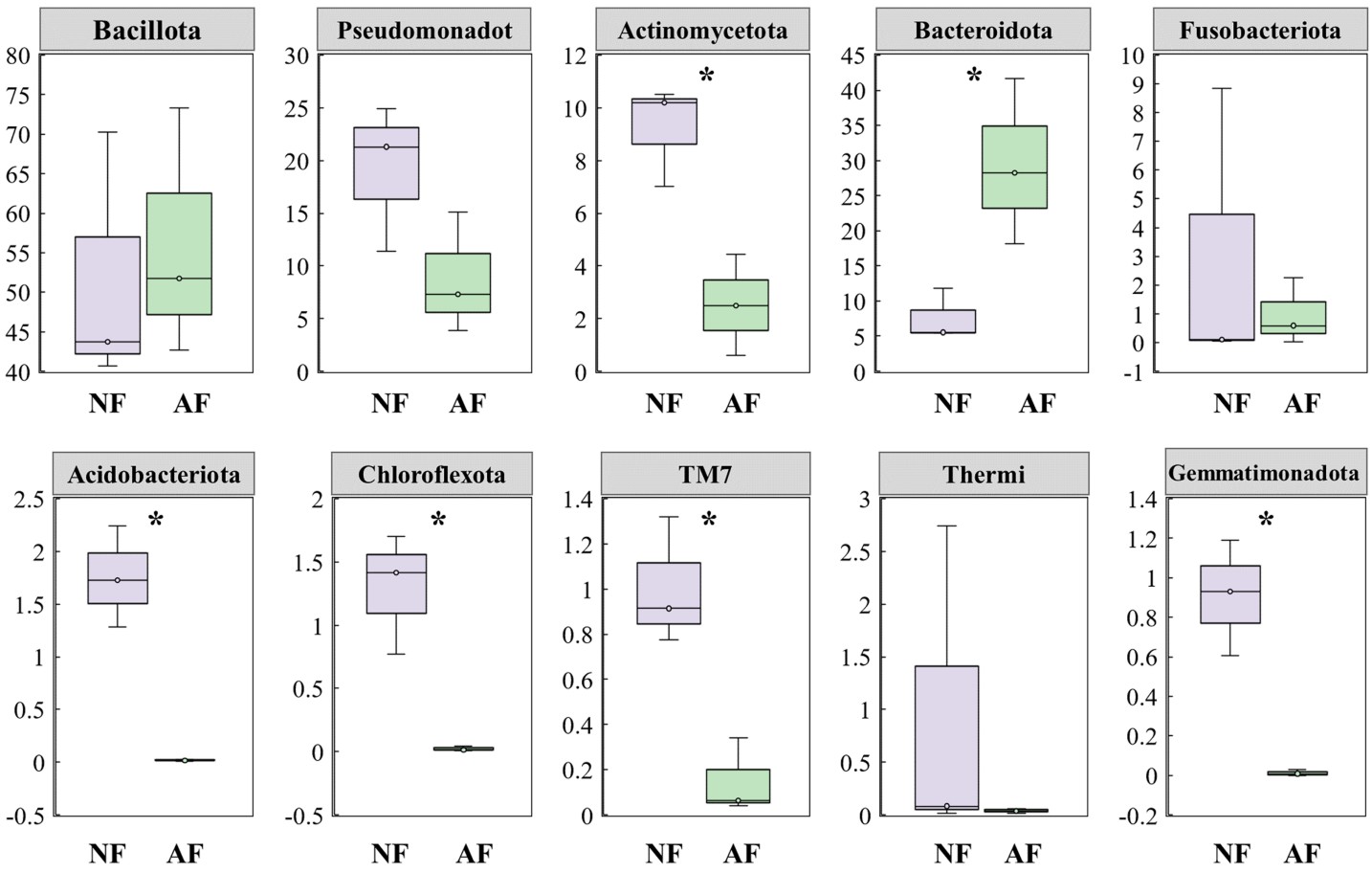

**Figure 4 Comparison of the top 10 dominant microbial taxa at the phylum level in mandarin fish fed with the FB and AF diets.** FB, fresh bait; AF, artificial feeds. The asterisk (*) at the top of the figures indicates significant differences between the two groups (*P* < 0.05).

metabolic activities related to lysosomal and sphingolipid metabolism were observed in the NF group.

## Carbohydrate and lipid metabolism related indicators

The carbohydrate and lipid related metabolic indicators were analyzed, as these pathways were highly enriched in the microbiota functional prediction. Photomicrographs of representative Oil Red O-stained liver sections from mandarin fish demonstrated that the AF group had significantly more lipid droplets than the NF group (Fig. 7A), which was also verified by the significantly lower number of lipid droplets (Fig. 7B; *P* < 0.05). In addition, the lipid content in the muscle of mandarin fish fed the AF diet was significantly higher than that of the NF group (Fig. 7C; *P* < 0.05). Based on the results of the Illumina mRNA-seq data, we further performed heat mapping and clustering of differentially expressed genes (DEmRNAs) associated with carbohydrate and lipid metabolism, including pathways of glycolysis/gluconeogenesis (fructose 1,6 bisphosphate aldolase (*fba2*), ATP-dependent 6-phosphofructokinase (*pfkm*), transketolase 10-like (*gapa2*), and glyceraldehyde-3-phosphate dehydrogenase (*gapdh*)), insulin signaling pathway (40S

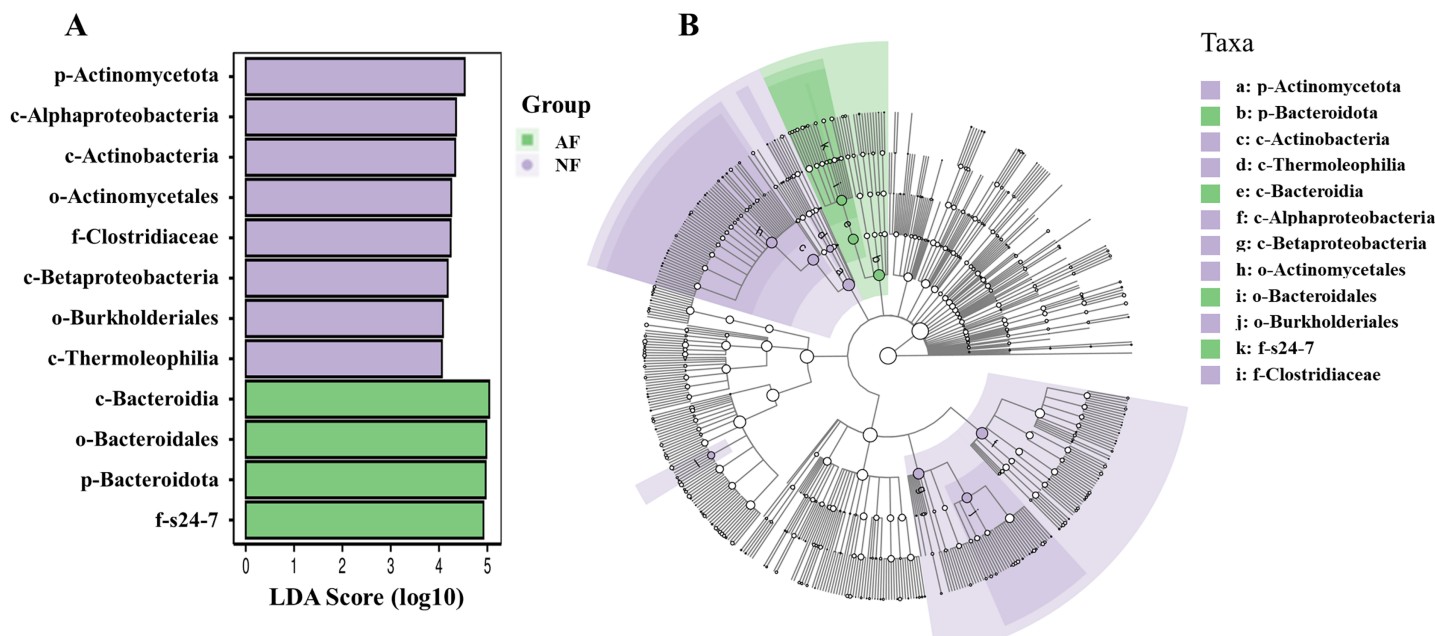

**Figure 5  Intergroup variation in the relative abundance of the intestinal microbial communities.** (A) Bacterial taxa differentially displayed in the gut of microbiota mandarin fish fed with the NF and AF diet. (B) Cladogram of LEfSe.               

ribosomal protein S6 (*rps6b*), uncharacterized protein LOC108887387 (*ppp1r3a*), parvalbumin-7 isoform X1 (*pvalb7*), and protein phosphatase 1 regulatory subunit 3C (*ppp1r3c*)), fatty acid biosynthesis (Beta-ketoacyl synthase (*ppsc*)), and PPAR signaling pathway (fatty acid-binding protein (*fabp7*), fatty acid desaturase (*fads2*), and fatty acid binding protein 11 (*pmp2*)) (Fig. 7D).

# DISCUSSION

## Loss of microbiota diversity in mandarin fish fed the AF diet

Diet has a significant impact on shaping and modulating the composition of the host gut microbiota (*De Filippo et al., 2010*; *Miclotte & Van de Wiele, 2020*). Generally, habitual dietary changes can disrupt the balance of the gut microbiota and have detrimental effects on overall health (*Qin et al., 2022*). In a sense, microbial diversity is an important indicator of gut function and is considered a marker of the healthy state of the intestinal microbiome (*Lozupone et al., 2012*; *Song et al., 2022*). In the present study, the richness was evaluated using the Chao1 and observed OTUs values, and the diversity of microbiota was evaluated using Simpson's index. The results demonstrated a significant reduction in both the richness and diversity of the gut microbiota in mandarin fish fed the AF diet, as reflected by the significantly lower levels of Chao1, Shannon, and observed OTUs according to the alpha diversity analysis. Similarly, the artificial diet feeding of silkworms (*Bombyx mori*), an insect with a feeding habit that feeds on natural foods such as mulberry leaves in the wild, resulted in a decrease in gut bacterial diversity and a simplified gut microbial structure (*Qin et al., 2022*). The loss of the microbial diversity can have detrimental effects, as it may create an environment that is more susceptible to pathogen invasion due to the

**A**

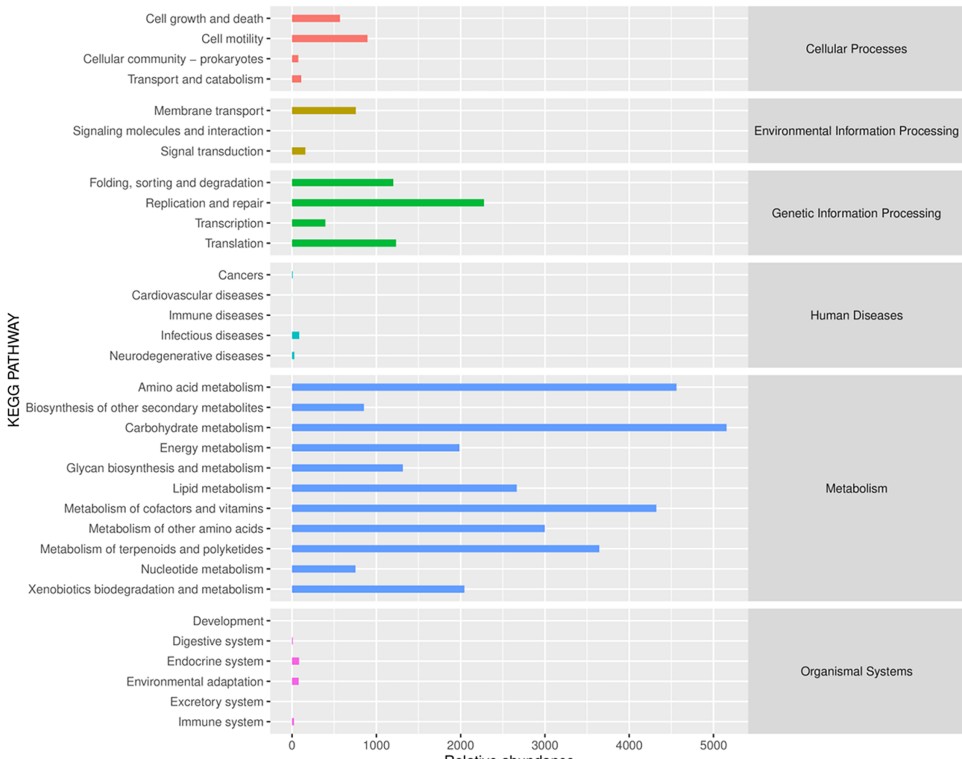

**B**

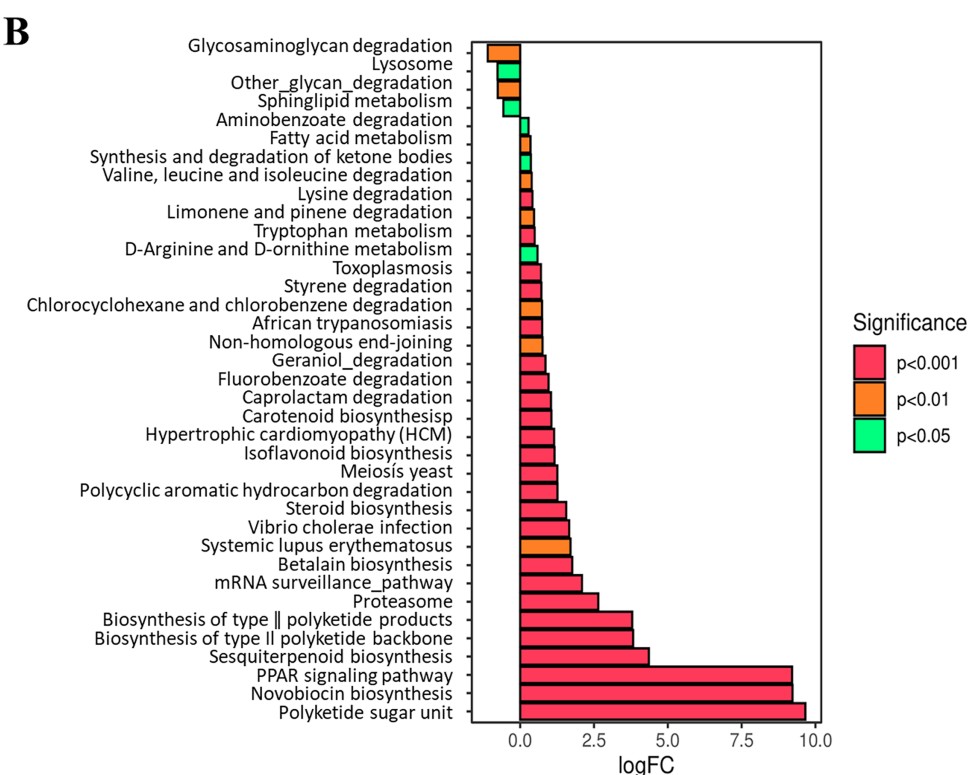

**Figure 6 Functional prediction analysis of gut microbiota in mandarin fish fed with the NF and AF diet using PICRUSt.** NF, natural feeds; AF, artificial feeds; PICRUSt, Phylogenetic Investigation of Communities by Reconstruction of Unobserved States. (A) Predicted functional categories annotated by
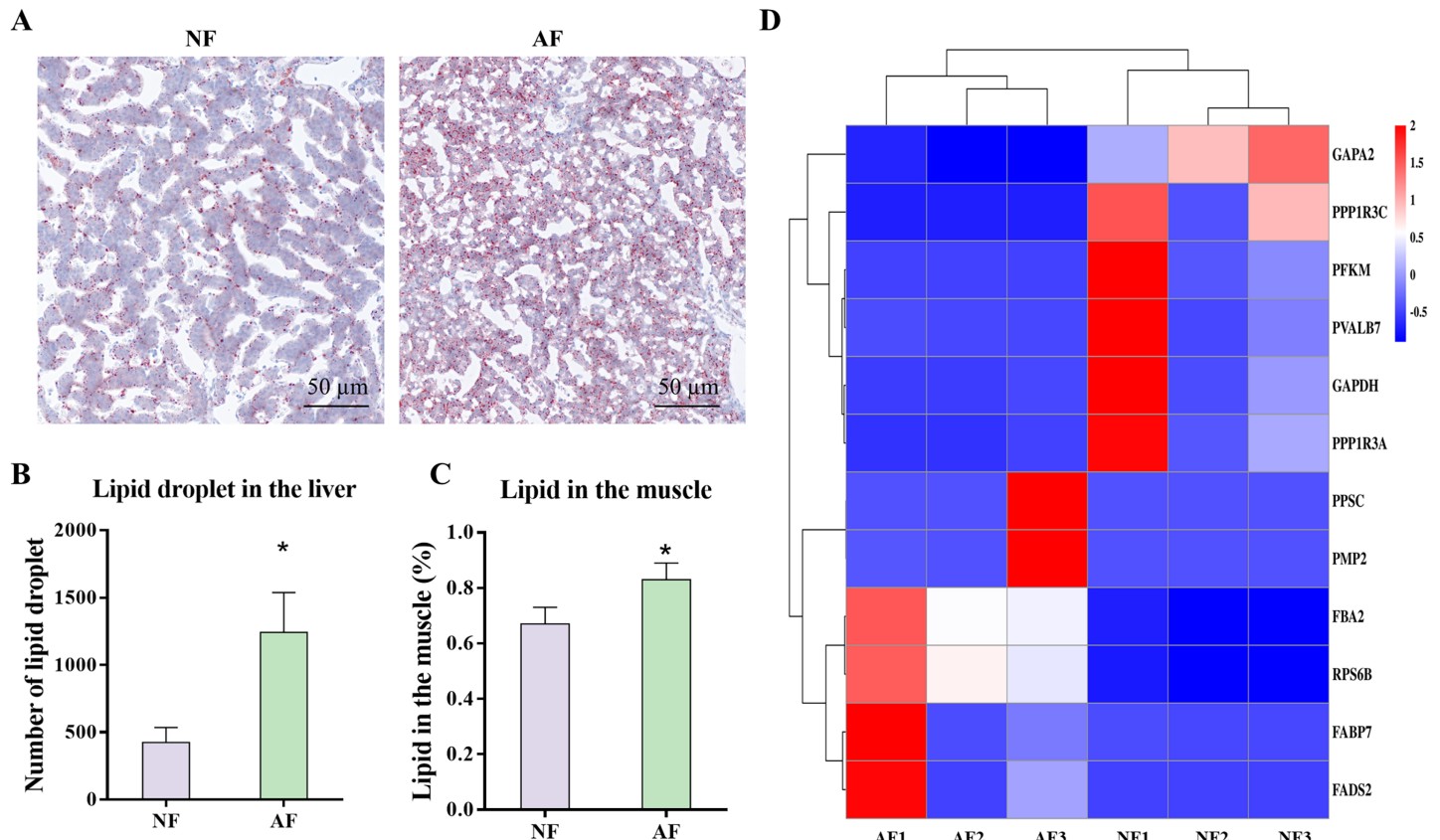

**Figure 6** (continued)
KEGG at Level 2. (B) Differentially enriched pathways based on KEGG annotation between the two groups of mandarin fish, with the NF as the control treatment.

**Figure 7 Gut microbiota-associated carbohydrate and lipid metabolism in mandarin fish fed with the NF and AF diet.** NF, natural feeds; AF, artifical feeds. (A) Oil red O staining of liver sections. (B) Number of lipid droplet. Lipid droplet was red-colored and nuclei was blue-colored. (C) Lipid content in the muscle. (D) Heatmap clustering of genes involved in carbohydrate and lipid metabolism in the mixed tissue of mandarin fish. Data represent mean ± SEM and are normalized to percentage of field area. The asterisk (*) at the top of the figures indicates significant differences between the two groups ($P < 0.05$).

missing of beneficial bacteria (*Schryver & Vadstein, 2014*). In general, the microbiome diversity of diseased individuals, including mammals and aquatic species, tends to be lower than that of healthy individuals (*Song et al., 2022*). Therefore, the decrease in richness and diversity in mandarin fish fed the AF diet could likely contribute to a compromised health status compared to those in the NF group, as evidenced by the mild intestinal damage and immune suppression in previous studies (*Li et al., 2023*).

The loss of microbiota in mandarin fish fed the AF diet can be attributed to several factors. One obvious aspect is that food not only provides organisms with nutrients for energy expenditure, but also serves as a source of colonizing microorganisms. The gut microbiota of fish includes microorganisms from the surrounding environment, including ingested food (*Cahill, 1990*). Compared to a mixed diet, a pure diet has been shown to

reduce the gut microbiota of fish (*Bolnick et al., 2014*). It has been documented that natural raw foods tend to contain a large number of resident bacteria and other microorganisms, while thermal processing methods like drying or extrusion can reduce the colonization of microorganisms in artificial diets (*Zhang & Li, 2018*). As a result, feeding the AF diet resulted in a reduction in the diversity and abundance of the gut microbiota in mandarin fish when compared to the NF diet, which can be attributed to the relative abundance of commensal microorganisms in the diet itself. Furthermore, the commensal microorganisms of the NF diet had the potential to reshape the host gut microbiota through interactions with the colonized microbes. Therefore, such interactions between the microorganisms in the diet and the host itself may increase the gut microbiota in mandarin fish fed with the NF diet. In addition, supplementing carbohydrates or plant-derived proteins to the diet can reshape the gut microbiota of animals (*Li et al., 2016*), and a high dietary carbohydrate may also reduce the microbial diversity in mandarin fish (*Zhang et al., 2020*). Moreover, our previous publication has shown that slight damage was found in the intestine of mandarin fish fed the AF diet (*Li et al., 2023*). Taken together, the inclusion of plant-derived proteins or carbohydrates in the AF diets led to a decrease in both the gut microbial diversity and richness, which may be related to the unexpected health status of the mandarin fish. Therefore, the present study examined the gut microbiota in fish exposed to NF and AF diet to mimic the evolutionary history of humans and to explore how these microbial communities have evolved over time in response to different dietary habits. By considering the adaptive mechanisms that have shaped gut microbiota composition, we can draw parallels between fish and human evolution in relation to food processing and its impact on microbiome diversity.

## The AF diet feeding remodulated the microbial structure of mandarin fish

By modifying the gut microbial community in fish, dietary composition causes biological changes that impact the population size of key symbiotic species and associated metabolism in the host (*Ringø et al., 2006*). The present results revealed that the AF diet remodulated the microbiota assemblages by altering their structure and composition. In terms of microbiota composition, the dominant OTUs were classified as Bacillota, Pseudomonadota, followed by Actinomycetota and Bacteroidota in both treatments of mandarin fish. This is consistent with previous studies in other aquatic species, such as totoaba (*Totoaba macdonaldi*), largemouth bass (*Micropterus salmoides*), and red drum (*Sciaenops ocellatus*), where Pseudomonadota and Bacillota were found to be the dominant core flora (*Yamamoto et al., 2021*; *Zhou et al., 2021*; *Larios-Soriano et al., 2022*). At the genus level, the relative abundance of the probiotic bacterium Lactobacillus was found to be relatively dominant in both treatments of mandarin fish. The PCoA analysis of the beta diversity revealed a clear separation between AF and NF treatments, suggesting that the gut microbiota structure in mandarin fish was distinct between the two dietary treatments.

Statistical analysis of relative taxa abundance at the phylum level showed that the AF group had a significantly higher relative abundance of Bacteroidota compared to the NF group. Bacteroides and Bacillota are two phyla of bacteria known for involvement in the fermentation and assimilation of nutrients in the host (*Wexler, 2007*). Bacillota have been associated with enhanced fatty acid absorption, thereby facilitating energy harvesting processes (*Ouyang et al., 2023*), while Bacteroidota are mainly important for the breakdown of high molecular weight organic material, suggesting their involvement in carbohydrate metabolism and energy metabolism (*Sidhu et al., 2023*). Moreover, studies have shown a positive correlation between a plant-based diet and the abundance of Bacteroidota in the gut, in that adopting plant-based diets has been associated with an increase in beneficial bacteria including Bacteroidota (*Sidhu et al., 2023*). In line with these findings, the present study revealed a significantly higher abundance of Bacteroidota in mandarin fish fed the AF diet, as supported by the bacterial composition and LEfSe results. Therefore, the inclusion of dietary carbohydrate in the AF diet may enhance the host's ability to assimilate carbohydrate and harvest energy by modulating the Bacteroidota in the gut of mandarin fish.

In contrast, significantly lower abundances of Actinomycetota, Acidobacteriota and Chloroflexota were found in mandarin fish of the AF diet treatment. Among these taxa, Acidobacteriota is associated with various functions such as carbohydrate metabolism, nitrogen metabolism, exopolysaccharide production, and transporter functions (*Kielak et al., 2016*). It has been reported that the relative abundance of Acidobacteriota was significant lower in hybrid grouper fed a formulated diet compared to those fed with chilled trash fish (*Ye et al., 2020*). This finding aligns with our results, which also indicate a lower occurrence of Acidobacteriota in the AF group of mandarin fish. In addition, Actinomycetota have been recognized as pivotal biomarkers in maintaining gut homeostasis by producing beneficial secondary metabolites and reducing the pathogenicity of pathogens (*Kielak et al., 2016*). Actinomycetota have also been reported to be implicated in modulating gut permeability and the immune system, including modulating immune-inflammatory and autoimmune responses by inducing regulatory T-cells (*Binda et al., 2018*). The significantly lower relative abundance of Actinomycetota observed in mandarin fish fed the AF diet suggested that these fish may have an inferior immune response compared to those in the NF group. This prediction is consistent with our transcriptome analysis results between the two treatments, which showed that the NF group had significantly higher expression of genes associated with TGFβ signaling and tight junctions, indicating enhanced immune and gut barrier functions (*Li et al., 2023*). Similar findings have been reported in other fish species, such as the freshwater drum (*Aplodinotus grunniens*), which showed a significant down-regulation of *Actinobacteriota* and *Chloroflexi* in the gut when fed an artificial diet (*Lozupone et al., 2012*). Overall, the consumption of the AF diet reshapes the composition and dominant bacterial taxa in mandarin fish, implying the potential effects of processed food on remodeling the host microbial structure.

## Gut microbiota-associated nutrient assimilation in mandarin fish fed the NF and AF diets

The intestinal tract is the main site of nutrient assimilation, not only through the functions of the intestinal villi in nutrient absorption, but also through the involvement of the gut microbiota in energy production from nutrient biotransformation. According to the predictive functional profiles of microbial communities using PICRUSt analysis, pathways related to the carbohydrate metabolism, amino acid metabolism, and lipid metabolism were highly enriched in mandarin fish. Moreover, mandarin fish fed the AF diet showed significantly higher metabolic activities in PPAR signaling pathways and steroid biosynthesis, which aligns with our previous findings from the integrated miRNA-mRNA analysis of mandarin fish (*Li et al., 2023*). Similarly, improved nutrient utilization and cellular homeostasis were observed in the gut of freshwater drum (*Aplodinotus grunniens*) under feed domestication (*Song et al., 2022*). In addition, it has also been reported that processed foods in the human diet often contain high levels of sugar and other additives, which could potentially increase nutrient absorption as well as the risk of type 2 diabetes (*Cuevas-Sierra et al., 2021*; *Fardet, 2016*). Therefore, the AF diet enhanced nutrient assimilation activities in mandarin fish, suggesting that variations between the natural and processed food caused differences in nutrient assimilation in the host. Overall, by investigating how gut microbiota contribute to nutrient metabolism and overall health in mandarin fish, we can gain potential insights into the consequences of consuming processed *vs* natural foods on host physiology in vertebrates.

To validate the results of the PICRUSt analysis, the nutrient assimilation of the gut microbiota was assessed by measuring the lipid levels in the liver and muscle and by analyzing the expression of gene related to carbohydrate and lipid, based on the transcriptome analysis. Significantly higher levels of lipid droplets in the liver sections and increased lipid content in muscle indicated that the AF diet induced significantly higher lipid deposition compared to the NF diet. This increased lipid deposition was correlated with the significantly higher expression of genes involved in the PPAR signaling pathway, including *fads2*, *fabp7*, and *pmp2*, as shown in the heatmap. In general, *fabp7* is an essential fatty acid binding protein that plays a crucial role in facilitating lipid transport, while *fads2* is involved in the desaturation process of converting 18:2n-6 to 18:3n-6 during fatty acid elongation (*Huang et al., 2022*). Consequently, the active molecular events observed in *fabp7* and *fads2* in the mandarin fish fed with the AF diet are also consistent with the metabolic changes and microbiota adaptations observed in the study. In addition, a significantly higher abundance of Bacteroidota was observed in the AF group. Bacteroidota have been reported to be positively correlated with the consumption of industrially processed food in humans (*Cuevas-Sierra et al., 2021*). On the other hand, Lactobacillus is a probiotic fermenting organism that can produce lactic acid and acetic acid during the fermentation of plant protein and energy sources (*Zhang et al., 2022*). The increase in the abundance of these two taxa aligns with the improved ability of fermentation and nutrient assimilation in mandarin fish fed the AF diet. Therefore, the inclusion of carbohydrates and plant-derived protein in the AF diet reshaped the gut microbiota, resulting in an

increased proportion of Bacteroidota. In conclusion, the changes in the composition and related functions of the gut microbiome, together with the altered molecular metabolic events, collectively contributed to the enhancement of fermentation and nutrient assimilation processes in mandarin fish fed the AF diet. From a limited point of perspective, the results highlight the potential impact of dietary changes on nutrient assimilation and host metabolism.

## CONCLUSIONS

As fish species are one of the most representative vertebrates in our biosphere, mechanisms towards mammals could be revealed by insights from fish vertebrates. Therefore, the present study examined the long-term feeding of AF and NF diets to mandarin fish to emulate the consequences of consuming processed and natural foods on gut microbiota assemblages in the host. The findings demonstrated that feeding with the AF diet reduced the abundance and diversity of gut microbes to some extent. In addition, Bacteroidota was found to be the biomarker in the gut of mandarin fish fed the AF diet, while Acidobacteriota was distinguished in the NF treatments. The AF diet feeding showed improved nutrient assimilation, which was supported by the composition and related functions of the gut microbiota, as well as the enrichment of active molecular events. As a whole, it is acknowledged that the short-term modulation on the microbiota subjected to the NF and AF diets could not fully reveal the mechanism concerning the human microbiota evolution during long period of time, the authors hope that the results of the present study could provide some potential insights for the research on the evolution of the microbiota in response to dietary variations in animals and humans to some extent.

## ACKNOWLEDGEMENTS

The authors would like to thank LC Sciences (Hangzhou, China) for their technical assistance.

### Funding

This research was funded by the Guangdong Basic and Applied Basic Research Foundation (2023A1515010008; 2021A1515111077), the Key Laboratory of Breeding Biotechnology and Sustainable Aquaculture, the Chinese Academy of Sciences (2023FB02), the Science and Technology Program of Guangzhou (2024A04J4584), the Scientific Innovation Fund, PRFRI (2023CXYC2, 2024CXYC1), and the Central Public-Interest Scientific Institution Basal Research Fund, CAFS (2023TD62). The funders had no role in study design, data collection and analysis, decision to publish, or preparation of the manuscript.

### Grant Disclosures

The following grant information was disclosed by the authors:
Guangdong Basic and Applied Basic Research Foundation: 2023A1515010008; 2021A1515111077.

Key Laboratory of Breeding Biotechnology and Sustainable Aquaculture, Chinese Academy of Sciences: 2023FB02.
Science and Technology Program of Guangzhou: 2024A04J4584.
Scientific Innovation Fund, PRFRI: 2023CXYC2, 2024CXYC1.
Central Public-Interest Scientific Institution Basal Research Fund, CAFS: 2023TD62.

## Competing Interests

The authors declare that they have no competing interests.

## Author Contributions

- Hongyan Li analyzed the data, authored or reviewed drafts of the article, and approved the final draft.
- Shuhui Niu performed the experiments, prepared figures and/or tables, and approved the final draft.
- Houjun Pan conceived and designed the experiments, authored or reviewed drafts of the article, and approved the final draft.
- Guangjun Wang conceived and designed the experiments, prepared figures and/or tables, and approved the final draft.
- Jun Xie conceived and designed the experiments, authored or reviewed drafts of the article, and approved the final draft.
- Jingjing Tian analyzed the data, prepared figures and/or tables, and approved the final draft.
- Kai Zhang performed the experiments, prepared figures and/or tables, and approved the final draft.
- Yun Xia performed the experiments, prepared figures and/or tables, and approved the final draft.
- Zhifei Li performed the experiments, prepared figures and/or tables, and approved the final draft.
- Ermeng Yu analyzed the data, prepared figures and/or tables, and approved the final draft.
- Wenping Xie analyzed the data, prepared figures and/or tables, and approved the final draft.
- Wangbao Gong conceived and designed the experiments, authored or reviewed drafts of the article, and approved the final draft.

## Animal Ethics

The following information was supplied relating to ethical approvals (*i.e.*, approving body and any reference numbers):

The study was conducted strictly according to the guidelines of the Institutional Animal Care and Use Ethics Committee of the Chinese Academy of Fishery Sciences (No. LAEC-PRFRI-2021-01-01).

## DNA Deposition

The following information was supplied regarding the deposition of DNA sequences:

The dataset generated by deep sequencing platforms is available at the NCBI Sequence Read Archive: PRJNA897818.

## Data Availability

The raw measurements are available in the Supplemental Files.

## Supplemental Information

Supplemental information for this article can be found online at http://dx.doi.org/10.7717/peerj.17520#supplemental-information.

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
