# Peer review of "Modulation of the gut microbiota by processed food and natural food: evidence from the Siniperca chuatsi microbiome"

_PeerJ, doi:10.7717/peerj.17520_

## Round 0.1 · original submission · Major Revisions

Please provide a detailed point-by-point rebuttal letter to each of the reviewers' comments, along with your revised manuscript.

Reviewer 1 ·

Basic reporting

The overarching goal of the study is not clear. It is stated that the use of mandarin fish acts as a model for the effects of processed food in mammals, but that connection is not really described in the discussion. In addition, the microbiota of fish is constantly infused with bacteria from the surrounding water, the microbiota of which is likely also influenced by the chemical composition of the diets. It is stated in multiple places that AF are potentially harmful to health, but in many other places suggests an increase in beneficial bacterial groups and nutrient assimilation. This manuscript would benefit from providing clarity to these points.

Experimental design

The methods state that n=6 for each diet treatment (2 fish per pond per diet treatment), yet throughout the manuscript, there is only microbiota data represented for n=3 (Figure 2C-D, Figure 3 A-B). The analyses should include each individual fish, unless samples were pooled within a replicate pond which is not stated in the methods.

Validity of the findings

The discussion seems to go back and forth on AF being harmful (inflammation seen in previous study, reduced diversity) and beneficial (greater energy harvest). Since this cannot be determined from microbiota structure alone, it is important to know the growth and survival of fish in these treatments.

There are areas where the statistical analyses should be specified where they are not.

Additional comments

Below are all the recommended suggestions.

Overall Major Comments
The overarching goal of the study is not clear. It is stated that the use of mandarin fish acts as a model for the effects of processed food in mammals, but that connection is not really described in the discussion. In addition, the microbiota of fish is constantly infused with bacteria from the surrounding water, the microbiota of which is likely also influenced by the chemical composition of the diets. It is stated in multiple places that AF are potentially harmful to health, but in many other places suggests an increase in beneficial bacterial groups and nutrient assimilation. This manuscript would benefit from providing clarity to these points.

Throughout, natural foods are referred to as baits. This is misleading if the paper is referencing aquaculture. I suggest using natural feeds (NF) and artificial feeds (AF) and being consistent throughout. For example, artificial food is sometimes referred to as processed food, and natural food and sometimes referred to as fresh bait.

The methods state that n=6 for each diet treatment (2 fish per pond per diet treatment), yet throughout the manuscript, there is only microbiota data represented for n=3 (Figure 2C-D, Figure 3 A-B). The analyses should include each individual fish, unless samples were pooled within a replicate pond which is not stated in the methods.

Other Comments:
Abstract
• Lines 33-34: Suggest rephrasing as “indicated that the two diet treatments were associated with distinct bacterial communities” or something similar
• Line 36: Either define LEfSe and PICRUSt or remove this part of the sentence altogether for clarity.
• Line 39: Italicize Lactobacillus. Also, was it just one OTU identified as this genus or were there multiple that you are considering? If there are multiple, change sp. to spp.
Introduction
• Industrialization and alteration of the human diet versus those of primates has been occurring for centuries. In addition, humans and primates are different species, and it has been shown that there is a species effect on the microbiota composition. This study looks at short-term influence of diet changes, so it does not seem to answer this question.
• Lines 59-60: Correct grammar in this sentence
• Line 63: ‘remain’ should be ‘remains’
• Lines 66-67: Not sure what this sentence is trying to say. Rephrase.
• Lines 69-80: This paragraph needs to be restructured for clarity. Are you referring to wild fish species or cultured fish here? The first few sentences mention natural baits and life in the wild, which suggests fishing. Do you mean natural prey? The rest of the paragraph is dedicated to farming and domestication.
• Lines 75-76: Check for grammar. What do you mean by ‘practical’ rearing conditions?
• Line 82: remove ‘the’, i.e. “promote large-scale culture”
• Lines 88-89: Include citations for each fish species
• Lines 89-90: Rephrase. Perhaps “Therefore, the effects of artificial feeds on the gut microbiota of mandarin fish warrants further investigation.”
• Line 91” Remove ‘Therefore’
• Line 95: Only bacteria were analyzed, so suggest using ‘bacterial assemblages’
• Lines 95-96: whose metabolic changes?
Methods
• Line 113: Since this is an underlying principal of the study, specify which diet
• Lines 117-120: This study took place in outdoor ponds, which can vary in water quality parameters. Please provide analyses to determine if water quality (temp, DO, pH, ammonia, light) differed between treatments. Also, was the photoperiod truly 12:12?
• Sampling: Did you take final weights of the fish or determine mortality? If so, this data should be included and analyzed between treatments.
• Line 124: Remove ‘of’, i.e., “Two fish”. Same in line 126: “1.5 mL sterile tubes”
• Line 129: remove ‘with’, i.e. “were disinfected”
• Line 129: How long were the tools disinfected? Or did you use flame sterilization?
• Lines 131: “the surviving animals were euthanized”. Also, consider rephrasing “subjected to a harmless treatment process”
• Line 140-141: Remove “after checking the integrity by gel electrophoresis” as this is stated in the previous sentence
• Line 144: should read “in a 25 µL reaction, containing”
• Line 146: change ‘u’ to ‘µM’
• Line 147: should read “followed by 30 cycles”
• Lines 151-152: What parameters were considered acceptable for the Bioanalyzer?
• Line 155: What were qualified amplifications? Did any samples fail?
• Bioinformatics: Were any sequences removed based on size or other QC filtering (for example, chimeras)? Length distribution in your supplementary figure ranges from less than 250 to about 450. Since these are paired-end reads, the length should be at the higher end, and the shorter sequences are likely not contigs and should be removed.
• Line 175: Specify that the venn diagram shows shared and unique OTUs by diet
• Line 175: Define PERMANOVA
• Line 176: Consider using the term ‘treatment’ instead of ‘groups’ throughout the paper
• Line 177: What taxonomic level was used in the Lefse analysis?
• Line 180: What is meant by ‘high-cost power’?
• Line 197: Why did you choose to pool those three tissues?
Results
• Line 215-216: Suggest including rarefaction curves in supplementary figures
• Line 219: should read “fecal microbiota”, as it has been shown that the adherent intestinal microbiota is different than that of the digesta
• Alpha diversity: Change “observed species” to “observed OTUs” here, in methods, and in figure.
• Lines 227-229, Figure 2A: Is the rank abundance curve necessary? The alpha-diversity measures already indicate that the diversity is higher in FB
• Lines 229-232, Figure 2B: The venn diagram depicts unique and shared OTUs, so make sure this is consistent throughout. I would not use ‘core OTUs’ without specifying how you defined what were core OTUs. Simply being shared is not typically a metric for core members of the community.
• Lines 233-237, Figure 2 C&D: Where are the replicates here? If n=6, there should be six points for each diet treatment.
• Line 241: ‘phylum’ should be ‘phyla’
• Lines 245-247, Figure 3B: unidentified should not be italicized, and the _ should be removed between unidentified and the taxa names
• Lines 250-251, Figure 4: How were these phyla tested statistically? Was it with Lefse, T-tests, or something else?
• Line 257: Again it’s important to know what taxonomic level you used for Lefse. The results in this area are stated in terms of phyla, but all of that information has already been stated. It may be more helpful to run this analysis at a lower taxonomic level.
• Lines 261-262: Remove first sentence. This is already stated in the methods.
• Lines 262-263: The 6 pathways in this figure are Level 1 pathways. Also, are these pathways for all samples, regardless of treatment?
• Line 265: metagenome Seq analysis is not mentioned in the methods. Additionally these are looking at Level 3 functions which is not mentioned in the methods
• Lines 274-277: What statistical analysis did you use for this data?
• Line 278: DEmRNAs not necessary to include since it’s never used again
• Carbohydrate and lipid metabolism related indicators: Specify patterns here, don’t just list the genes

Discussion
• Throughout: The discussion seems to go back and forth on AF being harmful (inflammation seen in previous study, reduced diversity) and beneficial (greater energy harvest). Since you can’t determine this from microbiota structure alone, I am curious as to the growth and survival of these treatments.
• Line 288: remove ‘abundance and’
• Line 294 & 296 & 305: ‘microbiota abundance’ should be ‘richness’
• Line 295: What about the other indices you measured?
• Line 302: how does reduced diversity create an environment favorable for pathogens?
• Line 304: remove ‘in’ after healthy individuals
• Line 306-307: Is dysbiosis a cause or effect?
• Line 320: Wouldn’t AF also interact with colonized microbes?
• Line 325: Do you have data on the carbohydrate content of these feeds?
• Line 338-339: According to the heatmap, Lactobacillus is only positively correlated with 3 samples, 2 being from AF.
• Lines 343-344: Should Bacteroides be Bacteroidota?
• Lines 346-347: Grammar. “(Ouyang et al. 2023), while Bacteroidota”
• Line 409: It is not stated or shown that Lactobacillus is significantly higher in the AF treatment.


Figures
• All figures: I suggest using the colors for your treatments consistency. For example, in this figure and Fig 2, FB is green and AF is purple, whereas in Fig 1 and 5, AF is green and FB is purple.
• Figure 3: I recommend switching what you have for Fig 3B with the supplemental figure S2. It provides greater detail in the microbiota composition and may be more informative to the readers.
• Figure 5: With the way the results are written, this figure does not provide much more information than Figs 3 and 4 because only phyla are described. Perhaps some of these other taxonomic levels should be discussed?
• Figure 5A: I’m assuming that you are only presenting the Lefse results for taxa with an LDA > 4? If so that should be stated.
• Figure 5B: The letters here are difficult to read. Is there a way to make them more legible?
• Figure 6A: Is this all functions found in the fecal microbiota, regardless of diet?
• Figure 6B: Which diet is the ‘control’ group to which the other one is compared? This is important for understanding increases and decreases in predicted metabolic function. Also, remove _ from pathway names
• Figure 7B: Is this data for the liver? If so, specify in figure caption

·

Basic reporting

The manuscript is written in professional, unambiguous English, only, I would suggest rephrasing the title, as there is no “re-modulation” (which means a second modulation) of the microbiota, instead there is differential modulation/effect by the processed & natural food.

There is basic background provided, however, citations are very local and not very international, I think for example, that references are missing in lines 87-89. In general, I think it would be noteworthy to mention that lots of research has been performed in wild against cultured fish, and the principal factor on those comparisons modulating microbiota might be the food consumed from wild against artificial in culture, although other factors (i.e. temperature, pH, other environmental) are influencing these changes. The work presented here can be viewed as an experiment where a unique factor is evaluated: food. However, I think work is self-contained with relevant results to hypotheses.

The structure, figures and tables are clear, although figure colors are too light colors, specially 1,2, and 4.

Links to the raw data in NCBI are shared.

Experimental design

The research is original and I think within Aims and Scope of the journal.

The research question is well defined, relevant & meaningful. It is stated how research fills an identified knowledge gap.

The investigation is performed to a high technical & ethical standard.

Methods described with sufficient detail & information to replicate, I would just point out that along the text, OTUS are mentioned to be analyzed, however there is no mention which identity was used to create OTUs. In figure 2 ASV/OTUs are mentioned as the same, but they are not the same. I think ASV should be dropped if the analysis was indeed performed with OTUs.

Validity of the findings

All underlying data have been provided; they are robust, statistically sound, & controlled.

Conclusions are well stated, linked to original research question & limited to supporting results. My only observation would be that the chemical composition of the diets is very much different (FB & AF). Normally the artificial diets are made after the fish requirements; however in this case protein and lipid content are far similar to FB profile, wouldn't that be a bad diet since the beginning for the fish?...it would have to be explained somewhere in the text.

Additional comments

Some lines and figures suggestions:

59-60 - the phrase is duplicated:..."has been strongly linked to has been
strongly related to"...
62-65 - it seems this is the work you cited before (De Filippo et al 2010)
84 - in --> of
129 - were with disinfected —> were disinfected
132 - duplicated with 134
294 - microbiota abundance —> microbial relative abundance
316-317 - rephrase “ diversity and richness” as “richness” is contained in the term “diversity”
318 - phrase is not clear when mentioning “resident” microorganisms

Figure 3. Although the classification of bacteria have just changed, use in the paper is up to date, however there are: Thermi and Acidobacteria in figure 3A that should be modified. And Figure 3B is not only genera but there are some family names.

Figure 6 - It is not clear in Fig 6B in which treatment expressed pathways are shown differencially.

---

## Round 0.2 · Minor Revisions

Please provide a detailed point-by-point rebuttal letter to each of the reviewers' comments, along with your revised manuscript.

Reviewer 1 ·

Basic reporting

I wish to commend the authors for addressing the previous comments, greatly improving the clarity of the submitted manuscript. There are still a number of grammar and spelling errors throughout that need to be corrected. The background and literature is sufficient. There are still a few items in the figures I would like to see resolved.
• Figure 1: Are your p-values all really equal to 0.05? Include the actual p-values for your tests here.
• Figure 2. Remove ASV from figure caption.
• Figure 7B: Specify in the caption or in the figure title that the lipid droplet data is from the liver.

Experimental design

There are a few areas where the experimental design/methods could be improved.
• Lines 108-109: What does ‘well-domesticated’ mean? Perhaps just say ‘domesticated’
• Lines 120-122: Thank you for showing the statistics on this in your revision. Please add a statement in the methods stating that these parameters did not differ significantly between treatments. Also, I recommend including the overall average survival and a statement that it did not differ between treatments.
• Bioinformatics section: Be sure to state any sequence filtering that occurred prior to data analysis (these are included in Table S2).
• Diversity and statistical analysis section: since you discuss good’s coverage and rarefaction curves in the results, please include that here in your analysis section.
• Lines 197-198: Were these samples pooled like the microbiota ones?

Validity of the findings

There are some areas of the discussion that could be clarified to strengthen and support stated conclusions.
• Lines 301-302: Thank you for providing the explanation to how reduced diversity creates a favorable environment for pathogens in your response. I suggest including this statement, including the citation, in the text of the manuscript.
• Line 325: I meant, do you have the carbohydrate concentrations in the diets you used in this study? It would be great to show that the carbohydrate concentrations are higher in AF if you are making this claim.
• Lines 326-327: What is meant by ‘unexpected health status’? Can you specify which data indicates that the fish fed the AF diets were not healthy and why?
• Lines 417: Without statistical analysis, you can’t say that Lactobacillus is ‘significantly’ higher. Either run the stats and show that in the results and figures, or remove it.
• Line 409: It is not stated or shown that Lactobacillus is significantly higher in the AF treatment.

Additional comments

Just a few additional comments.
Abstract
• Line 39: Lactobacillus is a genus, whereas Lactobacilli is a group. So this should either read “Lactobacillus spp.” OR “Lactobacilli”
Introduction
• Line 73: Rephrase “could exclusively accepted”, it’s unclear what this means
• Lines 97-98: suggest “alterations in bacterial assemblages and host-associated metabolic changes were evaluated in mandarin fish”

·

Basic reporting

I think this work is valuable as it contains metagenomics and transcriptomic data put together to uncover the effects of habitual dietary changes on the succession of the host microbiota in fish. However, I do not think this has potential insights into the knowledge of evolution because of the substitution of natural foods by processed foods in mammals, and in particular in humans. I think all evolution idea should be drop off. There is lots of work in evolution of human microbiota that would have to be revised but I do not think is the focus of the paper.

The authors have addressed my observations however the paper reads now a bit patchy, specially in the conclusions.

Experimental design

The authors have addressed my concern about the diet and the OTUs, but not completely, they do not mentioned OTUs identity used value yet. Also, they do not mention how many data from transcriptome they produced, and why didn’t they performed metatranscriptomics?

Validity of the findings

I do not know if it is the difficulty of the analysis, the interpretation, or the lack of differential results which reduces the data presented to the minimal, and I think this should be at least mentioned. It is a study with metagenomics and transcriptomics, which is interesting to see if analysis like these are worth it, however, in this case it feels results do not take us far and then conclusions are poor, and patched with last corrections.

---

## Round 0.3 · accepted · Accept

Thank you for the last revision, which has been satisfactory.